# Ocean forecasting of mesoscale features can deteriorate by increasing model resolution towards the submesoscale

Paul A. Sandery[1] & Pavel Sakov[1]

Submesoscale dynamics are ubiquitous in the ocean and important in the variability of physical, biological and chemical processes. Submesoscale resolving ocean models have been shown to improve representation of observed variability. We show through data assimilation experiments that a higher-resolution submesoscale permitting system does not match the skill of a lower resolution eddy resolving system in forecasting the mesoscale circulation. Predictability of the submesoscale is inherently lower and there is an inverse cascade in the kinetic energy spectrum that lowers the predictability of the mesoscale. A benefit of the higher-resolution system is the ability to include information content from observations to produce an analysis that can at times compare more favourably with remotely sensed satellite imagery. The implication of this work is that in practice, higher-resolution systems will provide analyses with enhanced spatial detail but will be less skilful at predicting the evolution of the mesoscale features.

---

[1] Bureau of Meteorology, 700 Collins Street, Docklands, 3008 VIC, Australia. Correspondence and requests for materials should be addressed to P.A.S. (email: paul.sandery@bom.gov.au)

The dynamics of the ocean at horizontal spatial scales $\mathcal{O}(1\,\text{km})$ and timescales $\mathcal{O}(1\,\text{day})$ are referred to as sub-mesoscale. This regime is considered to arise dynamically from mesoscale currents, is a form a quasigeostrophic turbulence and a pathway for energy dissipation towards finer scale mixing[1,2]. Submesoscale dynamics project onto a wide wave number spectrum and contribute to an inverse energy cascade towards the mesoscale, playing a role in exciting baroclinic instablilities[2,3]. At present, knowledge of the performance of increasing model resolution beyond mesoscale eddy resolving in ocean forecasting is limited. We investigate this by comparing 2.5 and 10 km horizontal resolution data assimilating ocean forecast systems of the East Australian Current (EAC) in the Tasman Sea. The EAC, like other Western Boundary Currents, is characterised by fast growing dynamical instabilities[4]. As a consequence relatively large forecast errors occur for eddy resolving ocean forecasting systems such as the Ocean Modelling, Analysis, and Prediction System (OceanMAPS)[5] and the Bluelink Reanalysis (BRAN)[6], which are able to mainly constrain mesoscale features to the available observations. Higher-resolution models should theoretically provide improved dynamical solutions as they resolve more and parameterise less of the sub-grid scale physics. When downscaling there remains, however, unresolved physics requiring recalibration of sub-grid scale parameterisations. Ocean modelling simulations of the Gulf Stream at submesoscale horizontal resolution have been shown to produce more realistic patterns than equivalent lower resolution simulations[7,8]. The characteristic spatio-temporal scales of submesoscale currents present challenges for the ocean observing system[1]. Currently, there are insufficient observations available to provide the necessary coverage required for submesoscale ocean forecasting. Therefore, ocean models which go beyond mesoscale eddy resolving cannot be sufficiently constrained or evaluated. The best we can do at present is understand whether the theoretical improvement from an increase in resolution provides a better forecast of the mesoscale. We investigate this by running 2.5 and 10 km horizontal-resolution regional ocean reanalyses with identical settings except for grid resolution.

Forecast innnovation errors (errors calculated in observation space) of the 2.5 and 10 km horizontal resolution systems indicate that the higher-resolution system is less skilful at evolving the mesoscale features of the ocean circulation. Qualitative comparisons made with remotely sensed satellite imagery and two additional reference reanalysis systems, BRAN3.5[6] and a 10 km resolution Ensemble Kalman Filter (EnKF) system[9] illustrate predictability in the higher-resolution system is particularly lower during fast growing dynamical instabilities. The results are presented in the next section, which is followed by a brief discussion and then a section describing the methods.

## Results

**Comparison of resolution.** The dimensionless Rossby number $Ro = U/fL$, where $U$ and $L$ are characteristic velocity and length scales and $f$ is the Coriolis parameter expresses the relative importance of Earth's rotation in a geophysical flow. The flow can be considered geostrophic when $Ro \approx 1$ and quasi to ageostrophic when $Ro > 1$. $Ro$ can also be defined as $\eta/f$, i.e., relative vorticity normalised by planetary vorticity. $\eta/f$ has been used to identify where advection dominates rotation and the presence of sub-mesoscale features in high resolution model fields[1]. Figure 1 shows $\eta/f$ for the 11 April 2006 for the two resolution systems. The 2.5 km resolution system contains partially resolved sub-mesoscale features in frontal areas between the mesoscale features

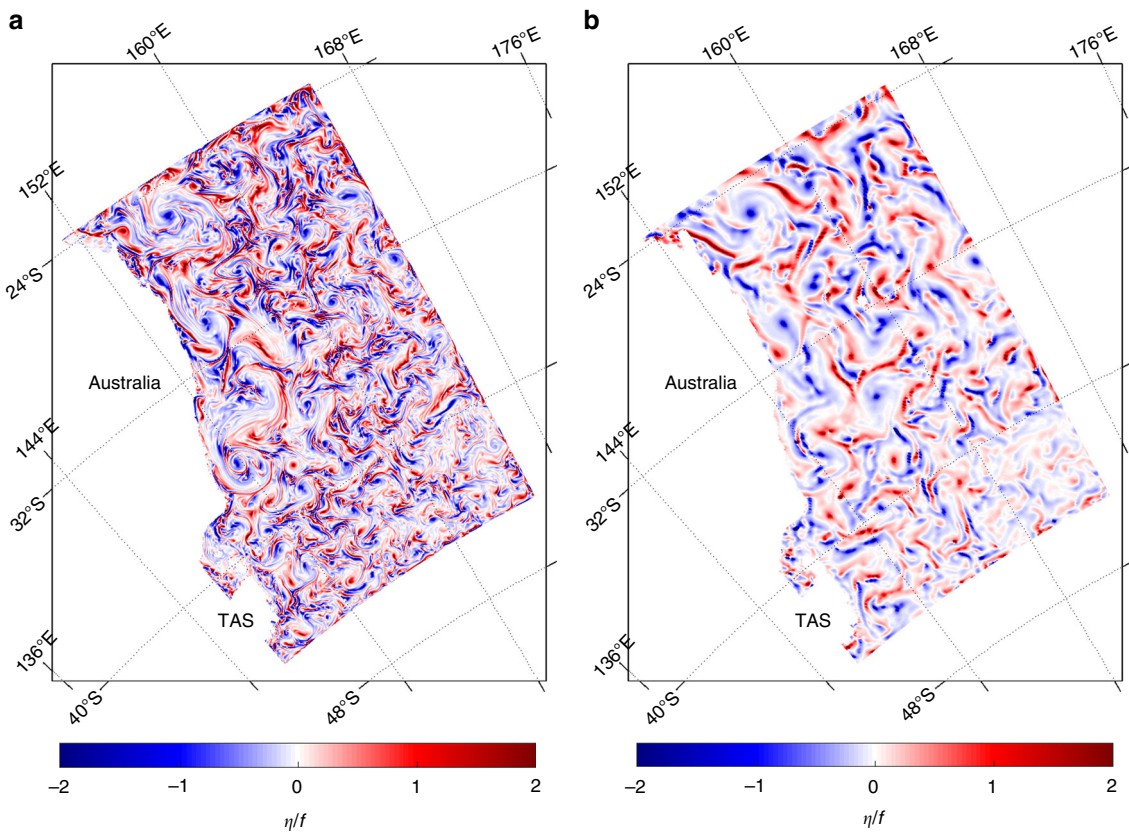

**Fig. 1** Comparison of resolution in the reanalysis systems on 11 April 2006. The metric shown is relative vorticity normalised by planetary vorticity ($\eta/f$). **a** 2.5 km resolution system and (**b**) 10 km resolution system. Tasmania (TAS)

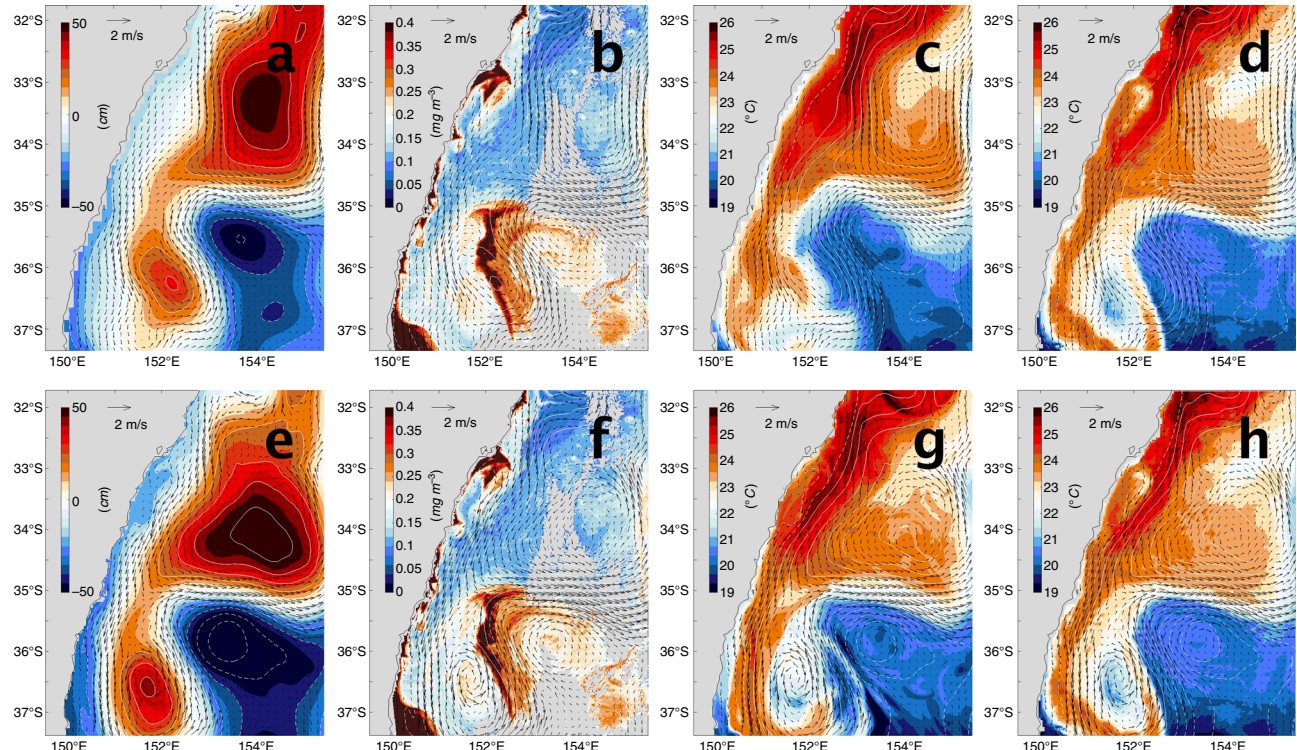

**Fig. 2** Analysis of mesoscale oceanographic circulation in the East Australian Current separation region on the 11th April 2006. **a** SLA[†] and SC[†], (**b**) MODIS chlorophyll-$\alpha$ with SLA[†] and SC[†], (**c**) SST[†], (**d**) AHVRR SST with SLA[†] and SC[†], (**e**) SLA[‡] and SC[‡], (**f**) MODIS chlorophyll-$\alpha$ with SLA[‡] and SC[‡], (**g**) SST[‡] and (**h**) AHVRR SST with SLA[‡] and SC[‡]. Sea level anomaly (SLA), Sea surface temperature (SST), Surface Current (SC), Advanced very-high resolution radiometer (AVHRR) and Moderate Resolution Imaging Spectroradiometer (MODIS). SLA contour interval is 10 cm. The thickened white contour represents zero SLA and dashed contours correspond to negative values. [†]10 km horizontal resolution system and [‡]2.5 km horizontal resolution system

**Table 1 Average forecast innovation statistics for 2006–2008 in the Tasman Sea region**

| Case | Description | SLA MAD (cm) | SLA bias (cm) | SST MAD (K) | SST bias (K) |
|---|---|---|---|---|---|
| 1 | 2.5 km nature run | 13.1 | −1.9 | 1.31 | −0.97 |
| 4 | 10 km nature run | 12.7 | 0.75 | 1.23 | −0.87 |
| 2 | 2.5 km reanalysis | 7.8 | −1.7 | 0.41 | −0.05 |
| 5 | 10 km reanalysis | 6.6 | 0.8 | 0.38 | −0.03 |
| 3 | 2.5 km upscaled | 7.7 | −1.5 | 0.40 | −0.04 |

SLA sea level anomaly, SST sea surface temperature, MAD mean absolute deviation, bias (mean error)
The cases are described in further detail in Table 4

where relatively large density gradients and baroclinic instabilities exist[2]. The 10 km resolution system permits quasigeostrophic dynamics in between mesoscale features, and on this day certain features of the mesoscale circulation are comparable between the two systems. This is the case as both are being constrained to the same observations. The 11 April 2006 is also an analysis day in the sequential analysis-forecast cycle. Figure 1 shows that the analysis of the 2.5 km system permits submesoscale dynamics and the model initialisation is not constraining the finer scales. The assimilation step in each cycle changes the mesoscale by small increments, allowing the higher resolution dynamics to continue to adjust and project onto each forecast in the sequence.

**Analysis of the East Australian Current**. A daily analysis of mesoscale oceanographic circulation in the EAC separation region on the 11th April 2006 for the 2.5 and 10 km systems is shown in Fig. 2. The images show that the EAC has generated an anticyclonic (AC) warm core eddy (positive sea level anomaly (SLA) feature in bottom left of panels). A cold-core eddy (circular area of negative SLA in bottom right of panels) is also present. The surface currents and SLA contours from both systems are shown over chlorophyll-$\alpha$ from the National Aeronautics and Space Administration (NASA) Moderate Resolution Imaging Spectroradiometer (MODIS) sensor on the Aqua satellite and Commonwealth Scientific and Industrial Research Organisation (CSIRO) 3-day composite advanced very-high resolution radio-meter (AVHRR) Sea Surface Temperature (SST) fields. MODIS chlorophyll-$\alpha$ data can be interpreted in the following way. Low concentrations represent warm southward flowing EAC water from the Coral Sea and AC eddies associated with positive SLA. High chlorophyll-$\alpha$ concentrations arise around fronts, in cold-core cyclonic eddies (areas of negative SLA) and in places of surface divergence, strong mixing and/or upwelling. It is clear that on this particular day the 2.5 km system is in much better agreement with the satellite images. This day is an analysis day in the sequential reanalysis, which means that observations for this time have been assimilated to adjust the model state. Figure 2 gives the impression that the 2.5 km system is a significant improvement compared to the 10 km system for describing the

**Table 2 Average sub-surface forecast innovation statistics for 2006–2008 in the Tasman Sea region**

| Case | Description | Temperature MAD (K) | Temperature bias (K) | Salinity MAD (psu) | Salinity bias (psu) |
|---|---|---|---|---|---|
| 2 | 2.5 km reanalysis | 0.59 | −0.23 | 0.093 | −0.014 |
| 5 | 10 km reanalysis | 0.56 | −0.20 | 0.090 | −0.006 |

Error metrics are mean absolute deviation (MAD) and bias (mean error)
The cases are described in further detail in Table 4

**Table 3 Average analysis innovation statistics for 2006–2008 in the Tasman Sea region**

| Case | Description | SLA MAD (cm) | SLA bias (cm) | SST MAD (K) | SST bias (K) |
|---|---|---|---|---|---|
| 2 | 2.5 km reanalysis | 3.6 | 0.5 | 0.18 | −0.01 |
| 5 | 10 km reanalysis | 3.5 | 0.3 | 0.18 | −0.01 |

SLA sea level anomaly, SST sea surface temperature, MAD mean absolute deviation, bias (mean error)
The cases are described in further detail in Table 4

mesoscale features in this region on this particular day, however, when we quantify the systems for their forecast error against all available observations from 2006-2008, we find that 11 April 2006 was a somewhat lucky day for the 2.5 km system.

**Quantitative forecast errors**. Average forecast innovation bias and mean absolute deviation for the reanalysis systems and their nature run counterparts from 2006–2008 are given in Table 1. These statistics are for the full Tasman Sea domain. The statistics in Table 1 for the nature runs illustrate the expected error of the systems when they are unconstrained. The 2.5 km system has larger unconstrained error than the 10 km system reflecting its larger variability. The statistics for the 10 and 2.5 km reanalysis systems show both are constrained by the observations. The average bias and mean absolute deviation for SST is comparable between the two systems, however, the same statistics for SLA show that the 2.5 km system has 1.2 cm higher average mean absolute deviation, which is significant. The upscaling of the fields of the 2.5 km system to 10 km resolution and the recalculation of the statistics using super observations, also at the 10 km scale, removes the smaller scale variability from both the higher-resolution model and the observations to make it statistically comparable to the 10 km system. These upscaled forecast innovations for the 2.5 km system show that the higher-resolution system has approximately similar mean absolute deviation to the 10 km system for SST, yet still over 1 cm error for SLA. Table 2 illustrates forecast errors for in-situ temperature and salinity. Here, the higher-resolution system has similar error to the 10 km system. Table 3 shows average analysis innovation error for the two systems and illustrates that the higher-resolution system is able to fit the observations at least as effectively as the 10 km system. Similar analysis error and larger average forecast error in the higher-resolution system, however, indicates that the error growth in the underlying mesoscale is more rapid, which has been permitted by the additional resolution.

**Qualitative assessment with satellite imagery and other reanalysis systems**. The additional resolved and unconstrained scales in the 2.5 km system generally lessen the predictability of the mesoscale compared to the 10 km resolution systems. Qualitative comparisons shown in Fig. 3 illustrate when and how this occurs. Here, we show daily mean SLA and SST from different reanalyses described in Table 4 and AVHRR SST and MODIS chlorophyll-$\alpha$ satellite images on days throughout 2007 when we found MODIS to provide reasonable coverage in the EAC separation region. This region is dominated by fast growing instabilities that drive formation of AC eddies, therefore all systems shown here exhibit differences due to the sometimes chaotic nature of the flow and the somewhat unconstrained nature of the systems - the sampling within the assimilated observations, particularly altimetric SLA, is limited in this region also. There are days such as 23rd May 2007 and 15th September 2007 where SLA fields from the four systems are in reasonable agreement with each other and with the MODIS chlorophyll-$\alpha$ data. Scanning across Fig. 3 shows on some days the 2.5 km system has improved respresentation of the details seen in the satellite imagery. When we look at the entire reanalysis this way, which is not shown here, we see that the 2.5 km system agrees less of the time with the satellite imagery compared to the 10 km systems. This is reflected in the average forecast error for SLA being ~1.2 cm larger in the 2.5 km system compared to the 10 km system (Table 1).

The 2.5 km resolution system evolves the mesoscale circulation less skilfully at times when fast growing instabilities are present. It is in these transition periods where the additional resolved dynamics project more onto the mesoscale. This can be seen in Fig. 3 for example on 23rd January 2007 where the separating EAC has become baroclinically unstable in the 2.5 km system to produce a warm westward transport into a region (approximately in the centre of the area shown) where other systems including MODIS data show the presence of a cold-core eddy. February to March 2007 is a transition period when a new AC eddy pinches off the EAC. The 2.5 km system appears to be less skilful at matching the MODIS pattern, however, after the transition, and after new data is assimilated the images for 12 April 2007 and 23 May 2007 show the improved agreement for the shape and position of the AC eddy and SST patterns compared to the other systems. This is arguably the case for 24 June 2007 and 3 July 2007 when the flow is also relatively coherent. Whilst there is also reasonable agreement for the remainder of the year, it can be seen that the 2.5 km system appears to initiate AC eddy formation earlier than its counterparts when there are minor further unstable flow transitions. Wave number eddy kinetic energy spectra of the two systems was compared (not shown) and we found the 2.5 km possesses elevated energy in the mesoscale wave numbers compared to the 10 km sytem. This indicates the higher-resolution dynamics project onto a wide wave number spectrum and that there is an inverse cascade. This is consistent with other reported findings[2] and supports our idea that predictability of the submesoscale projects onto the mesoscale.

**Discussion**

The mesoscale variability in a 2.5 km horizontal resolution regional model was constrained to observations using EnOI data

assimilation and compared to a 10 km horizontal resolution mesoscale eddy resolving system. Forecast errors for SLA, which infer the positions of the mesoscale features, are significantly larger in the higher-resolution system, reflecting less skilful representation of the evolving flow, particularly at times of fast growing instabilities. Analyses of the higher-resolution system, however, are able to provide improvements in describing detailed dynamical features observed in satellite remote sensing products,

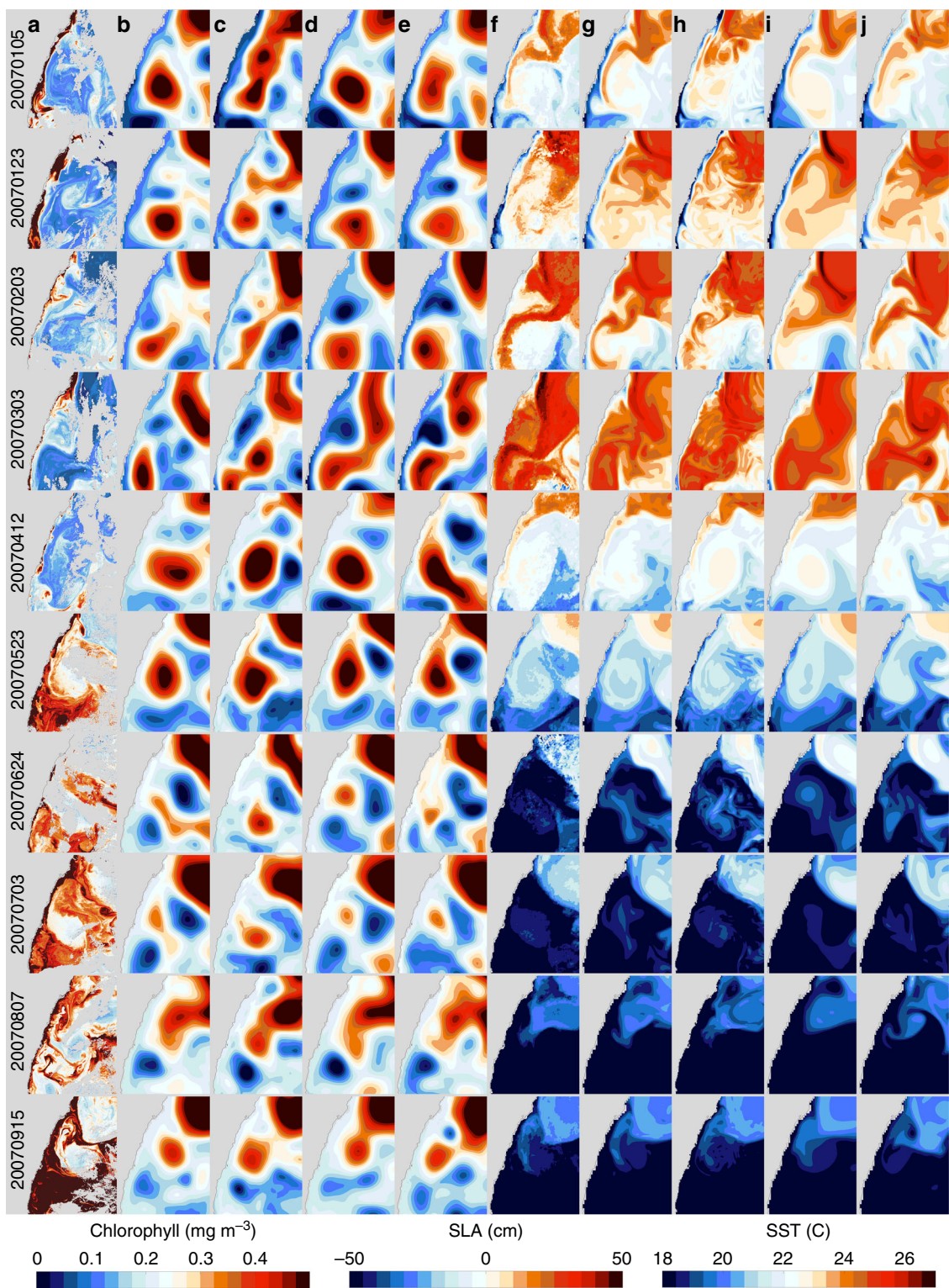

**Fig. 3** Daily SLA, SST, AVHRR SST and MODIS chlorophyll-α images on days throughout 2007 when MODIS provides reasonable coverage in the East Australian Current separation region. **a** MODIS chlorophyll-α concentration, (**b**) 10 km system SLA, (**c**) 2.5 km system SLA, (**d**) EnKF SLA, (**e**) BRAN SLA, (**f**) AVHRR SST, (**g**) 10 km system SST, (**h**) 2.5 km system SST, (**i**) EnKF SST and (**j**) BRAN SST. Sea level anomaly (SLA), sea surface temperature (SST), advanced very-high resolution radiometer (AVHRR), moderate resolution imaging spectroradiometer (MODIS), ensemble Kalman filter (EnKF), Bluelink reanalysis version 3.5 (BRAN)

**Table 4 List of datasets used in the study**

| Case | Horizontal resolution | Analysis method | Domain | Description |
|---|---|---|---|---|
| 1 | 2.5 km | EnOI | regional | nature run |
| 2 | 2.5 km | EnOI | regional | reanalysis |
| 3 | 2.5 km → 10 km | EnOI | regional | upscaled reanalysis |
| 4 | 10 km | EnOI | regional | nature run |
| 5 | 10 km | EnOI | regional | reanalysis |
| 6 | 10 km | EnKF | regional | reanalysis |
| 7 | 10 km | EnOI | global | BRAN3.5 reanalysis |
| 8 | 2 km | CA | regional | AVHRR SST analysis |
| 9 | 2 km | CA | regional | MODIS chlorophyl-$\alpha$ analysis |

EnOI Ensemble Optimal Interpolation, EnKF Ensemble Kalman Filter, CA composite analysis, BRAN3.5 Bluelink Reanalysis version 3.5, AVHRR advanced very-high resolution radiometer, SST sea surface temperature, MODIS moderate resolution imaging spectroradiometer
Cases 1–7 use ERA-Interim atmospheric forcing and Cases 1–6 are nested in BRAN3.5. The upscaled results are based on regridding the 2.5 km resolution forecast fields onto the 10 km grid and reprocessing the innovation statistics

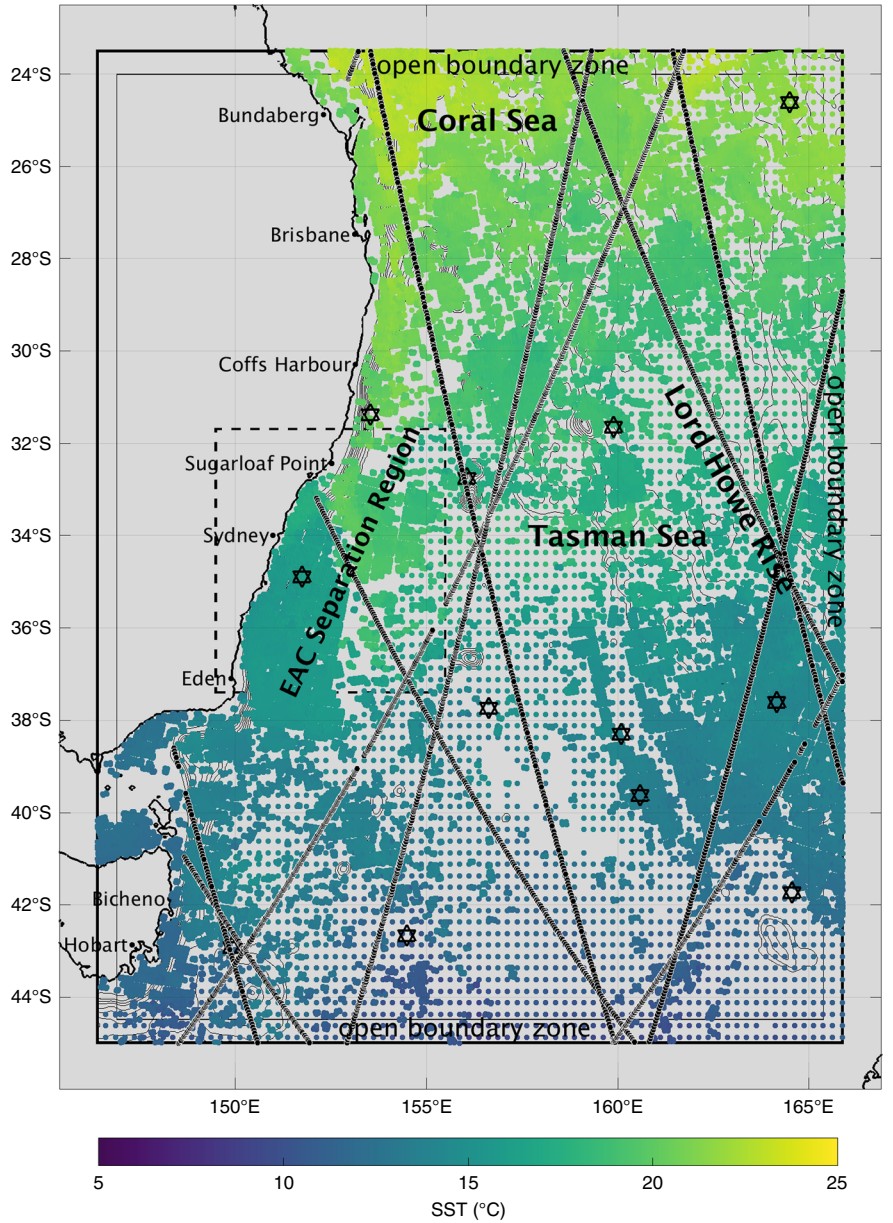

**Fig. 4** The study region. Also shown are areas delimited as open boundary zones, the East Australian Current (EAC) separation region and typical super-observation coverage for a single 3-day cycle for 14–16 September 2008. The coloured circles are sea surface temperature (SST), the black lines correspond to along track altimeter observations and the black stars are the positions of Argo profiles

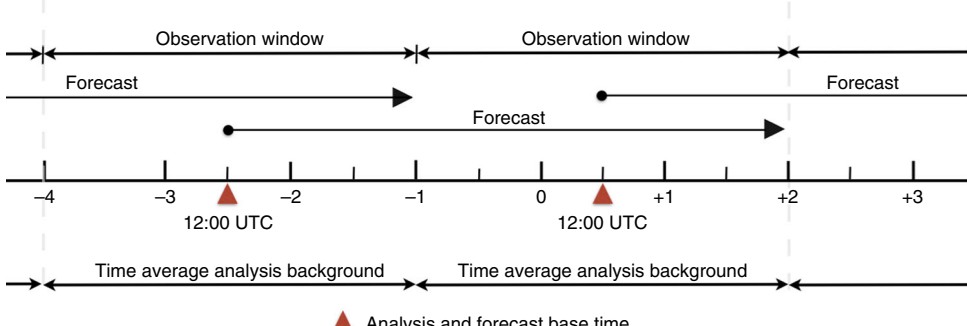

**Fig. 5** Analysis-Forecast cycle and forecast error estimation scheme. The regional reanalysis systems are run using a 3-day sequential cycle with a 3-day centred observation window. To calculate the increments, a time-averaged analysis background using the last 3 days of a 4.5 day forecast in each cycle is used to be consistent with the observation window. The increments are added to the instanteous model state at the analysis and forecast base time, which is taken to be 12:00 UTC. The forecast innovation errors are based on comparing the time-averaged forecast with the forward independent super observations in the 3-day window prior to assimilation in the next cycle. Super observations reduce representativeness error as they are spatially filtered by model grid scale and temporally filtered by the observation window

such as AVHRR SST and MODIS chlorophyll-$\alpha$ imagery, and at times are seen to be closer to the observed truth. The results are based on standard configurations of ocean model and data assimilation system. The two experiments utilise the same settings apart from differences in resolution. These results point to the existence of an intrinsic limit and the incapacity of a higher-resolution model to improve forecasting of the mesoscale circulation. The inverse cascade in the eddy kinetic energy spectrum is associated with communication of less predictable elements of the higher-resolution dynamics into the mesoscale and is considered to be the underlying reason for the larger forecast errors in the higher-resolution system.

The growth in computer power has enabled increases in model resolution and more realistic representation of dynamics and variability[8]. We found an increase in resolution doesn't automatically improve the skill of the forecasting system. Sub-mesoscale ocean forecasting certainly also requires better observation coverage, improved physics and sub-grid scale parameterisations.

## Methods

**Numerical model**. The systems use the three-dimensional primitive equation volume conserving ocean model MOM4p1[10] and standard bathymetric data typically used in ocean modelling[11]. The 10 km resolution grid is a regional cut out from the BRAN3.5 grid and the 2.5 km grid is a regridded version of this. The 10 km grid has 51 vertical levels and the top cell approximates quantities at 2.5 m depth with the average resolution in the upper 200 m being ~10 m. The 2.5 km model has 70 vertical levels with the top five levels being equivalent to the 10 km model, however, there is an increase in resolution between ~50 and 500 m. The geographic region for both models is the same and is within the coordinates 146.05°E to 165.95°E and 44.95°S to 23.05°S (Fig. 4). Also shown in Fig. 4 is the area delimited as the EAC separation region and typical super-observation coverage for the 2.5 km grid for a single 3-day cycle. The regional systems are nested in BRAN3.5 and use an adaptive relaxation scheme for open boundary conditions[12]. This is applied to sea level, three-dimensional currents, temperature and salinity in the open boundary zones shown in Fig. 4 similar to previous studies[4,9,12,13]. Both systems were forced by 3 hourly surface fluxes of momentum, heat and salt from the ERA-Interim atmospheric reanalysis dataset[14]. The physical model settings for the regional systems are identical, such as the use of a 4th-order Sweby advection method and a scale dependent isotropic Smagorinksy horizontal mixing scheme[15]. The latter is used to take care of recalibration of sub-grid scale mixing for the change in resolution. Vertical mixing is parameterized using the GOTM $\kappa$–$\varepsilon$ scheme. Note that explicit tides are not modelled in any of the systems, rather we use an implicit parameterised representation of tidal mixing[16].

**Data assimilation**. Ensemble Optimal Interpolation (EnOI)[17] is used to constrain the mesoscale features of both resolution systems to the available observations. We use a stationary ensemble of model error covariances derived from an 18 year nature run of an eddy resolving global model[6]. This provides both systems an analysis that is appropriate for the dominant spatio-temporal scales in the observations and filters higher wave number spatial scales from the analysis of the

higher-resolution system. Using covariances from the global model also allows continuity across open boundaries, which is important for long-term performance of the regional systems. Both systems are run with the same data assimilation settings as used in previous studies[9,13]. Data assimilating ocean reanalysis experiments are carried out over a three-year period from 2006 to 2008. The EnOI systems are run in a cycle scheme that centres and aligns a 3 day observation window with a time-averaged analysis background as shown in Fig. 5. The analysis increment for the state vector is added to the instantaneous model forecast state at the analysis time, i.e., the model is initialised directly to the analysis.

**Observations**. The assimilated observations are the same for both systems. Altimetric SLA data is taken from the Radar Altimeter Database System (RADS)[18] using tide, mean dynamic topography (MDT) and inverse barometer corrections. SLA observations are limited to water depths greater than 200 m. Model SLA is the difference between sea surface height and MDT from an 18 year nature run of the global model[6]. SST retrievals from the Naval Oceanographic Office (NAVO-CEANO)[19] and WindSat[20] databases are used. In situ temperature and salinity from Argo[21] and Conductivity Temperature Depth (CTD) profiles from CSIRO are also used. A super-observation is formed by combining all observations within a model grid cell into a single observation weighted by inverse error variance. The super-observation therefore places more emphasis on observations with lower error estimates. The super-observation also has a new error estimate that is lower than the original observations, thereby increasing the impact of the observations in the assimilation. This is standard practice in weather and ocean forecasting and reduces discrepancies related to representation error[22].

**Error quantification**. The performance of both systems is quantified in terms of forecast innovation error using all available forward and independent observations. A common innovation metric is root mean square deviation (RMSD), which can be dominated by a relatively small number of innovation elements with large magnitudes corresponding to either less observed or more chaotic parts of the model, or be influenced by observations with large errors as it does not account for observation error. Consequently it may not accurately represent the performance of the system. To minimise this we use Mean Absolute Deviation (MAD). MAD is a more robust metric than RMSD[23] and has been used to assess performance of ocean forecasting systems[9,13]. Regardless of the metric, the higher-resolution system carries a penalty in terms of innovation statistics compared to the lower resolution system, owing to its higher variability at smaller scales. This factor is minimised by re-calculating the innovation statistics of the 2.5 km forecast fields regridded back to the 10 km grid using second-order conservative remapping[24]. The statistics should therefore be comparable. The "upscaled" statistical analysis assesses whether the additional physics in the higher-resolution system contributes to improving forecast skill of the evolving mesoscale features.

**Code availability**. The ocean model is available at https://github.com/mom-ocean/MOM4p1 and the data assimilation code can be found at https://github.com/sakov/enkf-c. System and observation processing scripts are intellectual property of the Bureau of Meteorology.

**Data availability**. The data that support the findings of this study are available from the corresponding author upon reasonable request.

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

## Acknowledgements

This work was carried out within the Bluelink ocean forecasting project, a joint partnership between CSIRO, the Bureau of Meteorology and the Royal Australian Navy. High-performance computing was carried out at the National Computational Infrastructure (NCI) at the Australian National University (ANU) in Canberra, Australia.

## Author contributions

The two authors shared responsibility in designing the method, obtaining, analysing and interpreting the results and in writing the manuscript.

## Additional information

**Competing interests:** The authors declare no competing financial interests.

