## [Peer Review File · Nature Communications]

Reviewer #1 (Remarks to the Author):

Left comments for the Editor only.

Reviewer #2 (Remarks to the Author):

Sub-mesoscale ocean analysis and forecasting

Paul Sandery and Pavel Sakov

The paper examines the results of several Observation System Experiments to demonstrate the degree to which a high resolution numerical model may differ from a lower resolution result. The results are consistent with expectations. The higher resolution model provides more accurate results at scales that are resolved by the observation systems. There are a few points that would preclude acceptance of the paper at this time.

The writing is difficult to follow and understand. The first paragraph of section 3 seems to have lost sentences at the end. The discussion moves from Figure 3 to Figure 4 without pointing out that Figure 4 has become the focus, and the information described in Figure 4 is difficult to discern. I did not find a definition of the statistic MAD used in the presented tables. It is difficult to know what the results mean. Some additional statistics such as root mean squared error, conditional and unconditional bias, etc... would be quite valuable. These are issues that can be addressed easily with careful editing and attention to clear explanation.

Fundamentally there is a disagreement with what is submesoscale. The authors point out that the 10 km model is characterized by mesoscale dynamics and claim that the 2.5 km resolution model contains submesoscale. While the factor of 4 resolution certainly adds physical representation it does not cross from mesoscale to submesoscale. Formal convergence studies of mesoscale dynamics indicate that about 3 km resolution is the point at which numerical methods represent the differential equations involved in mesoscale eddy formation and propagation as measured by eddy kinetic energy levels (Hurlburt and Hogan, 2000). Convergence studies of submesoscale dynamics (Capet et al., 2008) indicate that 1 km resolution is the point at which submesoscale dynamics become represented by numerical methods. Figure 3 shows the vorticity relative to the local Coriolis parameter, and this ratio is a Rossby number. Mesoscale dynamics are characterized by small Rossby number, and submesoscale is reached when the Rossby number becomes order 1 or greater. From the colorbar scale, both these results indicate that the Rossby number is small. Therefore the flow is mainly geostrophically balanced and not reaching into the submesoscale. None of these considerations invalidate the results presented. These can easily be addressed by not referring to the 2.5 km model as submesoscale resolving, and the article title should be changed accordingly.

As computational capability continues to increase, numerical simulations are run at higher resolutions. The observational capability is not keeping pace with the Moore's law increase in computer power. Thus we are reaching the point where numerical models that represent instabilities in the ocean flow field are surpassing the ability of satellites and in situ data to provide information at the modeled scales. Models will generate realistic dynamics, they will generate realistic instabilities in the flow, these instabilities will not be synoptically placed compared to events in the real world, and thus even though numerical models have improved the results are counter intuitively degraded. This is a valuable understanding to bring to light. I think the authors can do this, though there are some issues as pointed out above that are presently obscuring the results and distracting from the main conclusion.

Hurlburt, Harley E., and Patrick J. Hogan. "Impact of 1/8 to 1/64 resolution on Gulf Stream model-data comparisons in basin-scale subtropical Atlantic Ocean models." *Dynamics of Atmospheres and Oceans* 32, no. 3 (2000): 283-329.

Capet, X. A. V. I. E. R., JAMES C. McWilliams, M. JEROEN Molemaker, and A. F. Shchepetkin. "Mesoscale to submesoscale transition in the California Current System. Part I: Flow structure, eddy flux, and observational tests." *Journal of Physical Oceanography* 38, no. 1 (2008): 29-43.

Reviewer #3 (Remarks to the Author):

Attached file "review.pdf" is PDF version of this review identical in content. Use this plain text version only as alternative.

Review of manuscript 'Sub-mesoscale ocean analysis and forecasting' by Paul Sandery and Pavel Sakov submitted into Nature Communications

Manuscript Number: NCOMMS-16-14750

Summary and recommendation

This manuscript deals with an emerging dilemma of using a relatively high-resolution model for the purpose of forecast and/or comparison with data available only at a coarser resolution. It is expected that grid refinement leads to improvement of model skill, however, it also results in a more turbulent and, correspondingly, less deterministic on the smaller scales solution which cannot be directly compared to the data (or constrained by the data in the case of data assimilation).

The approach adopted by the authors is to compare filtered model fields (in time in their case) and to use ensemble optimal interpolation which introduces some averaging of non-deterministic component of the solution.

The manuscript is complete -- although too short and somewhat formal (but this is specific to this journal) and well written. It can be published after correcting the items outlined below (some of which I believe are admitted by mistake), and I also suggest de-emphasizing word "sub-mesoscale" because as it used right now it artificially attached to the topic.

Detailed remarks

The use of term "sub-mesoscale" throughout the article is over-stretched and somewhat misleading. While there exist a range of definitions of what is considered to be submesoscale and what is not (or, more specific to the context of ocean modeling how fine the grid resolution should be for the results to be classified as sub-mesoscale), the common use of word "sub-mesoscale" (like sub-mesoscale processes, etc.) implies strongly ageostrophic dynamics, interactions vortex dynamics with surface boundary layer, formation of fronts and frontal instabilities, convergence zones, etc (c.f. Capet et. al. 2008). And, at the same time much larger scales than the ones at which full 3D turbulence may occur -- this would be the realm of LES modeling. Somewhat intuitive (but not uniformly accepted) criterion is the requirement that the boundary layer deformation radius should be well resolved by the horizontal grid.

There is an obvious analogy with the demand to resolve first baroclinic deformation radius for eddy-resolving model, however this analogy is somewhat superficial because the underlying processes are qualitatively different: baroclinic radius generally plays the role of limit to which the eddies are allowed to grow by mergers, absorptions, and axisymmetrizations -- it is believed that the underlying dynamics is essentially 2D or QG-type turbulence resulting in energy cascade from small to large scales; the sub-mesoscale dynamics is quite different. Besides boundary layer deformation radius is

not so easily define. However the de-facto consensus is that sub-mesoscale models require sub-kilometer resolution -- preferably a fraction of Δx .

What is actually going on in this study is a B-gridded model derived from MOM4p1 code run at $\Delta x = 2.5 \text{ km}$, introduced as (page 7, bottom) "To demonstrate that the 2.5 km system is sub-mesoscale eddy permitting and that the data assimilation and model initialization is not constraining ..." and compared against $\Delta x = 10 \text{ km}$ simulations using the same code. While $\Delta x = 2.5 \text{ km}$ is expected produce a healthy vortex dynamics, it is hardly capable to capture sub-mesoscale processes mentioned above. Especially if it is kept in mind that B-gridded MOM4p1 and MOM5 codes use second-order-accurate vector-invariant form (based on absolute vorticity and Bernoulli potential) for discretization of momentum equations. The use of B-grid is generally considered a kind of " $\sqrt{2}$ " disadvantage in comparison with a C-gridded model also using vector-invariant form and second-order discretization, e.g., MITgcm and NEMO [see, for example Griffies own Elements MOM (2012 release) Chapters 9 and 14; web link provided below in this review]. Furthermore, ROMS uses 3rd-order upstream-biased advection for the momentum equation, which is known to offer a significant advantage over the second-order numerics for advection of momenta (Lemarié et al., 2015; Soufflet et al., 2016 among others), which pushes threshold for submesoscale resolution for MOM4p1 code even further (it should be noted that most submesoscale simulations to date were produced by ROMS or by pseudo-spectral codes in idealized studies).

Nevertheless, this reviewer believes that the methodological approach of comparing 10 km and 2.5 km models for the purpose of making the point that "... whilst forecast errors are larger in the sub-mesoscale system, the system can nonetheless reproduce detailed dynamical features observed in satellite remote sensing products such as AVHRR SST and MODIS chlorophyll-imagery...' and...' is actually correct and worth making. The authors should just remove overemphasis on submesoscale (perhaps change wording of the title): after all what going from 10 to 2.5 Δx resolution results in solid improvement of model skill, and the methodology of comparing filtered model fields against the available data (rather than attempting a more direct comparison) is perfectly meaningful way to judge the model accuracy and errors.

page X-3, 1/3 from bottom, sentence "Higher-resolution models should theoretically provide improved dynamical solutions. This is because they resolve more and parameterize less of the subgrid-scale physics" -- actually this needs to be clarified or reformulates. Parameterization usually involves a physically-motivated component of the code designed to represent processes which ocean model cannot simulate directly due to lack of resolution or approximations made to fundamental equations or both. In principle this means that once grid resolution is refined, the modeler may choose to drop some of these components with the expectation that the processes they are designed to represent are now modeled directly. However, reading further into the article does not seem to support this -- what is going on there is just running exactly the same code at higher resolution. Please reformulate.

page X-5, 1/3 from bottom, reference [Griffes et al., 2009] -- the reference is essentially a manual, which is updated and expanded on regular basis. Why to refer to an outdated document when a newer and more

complete is available?

http://www.mom-ocean.org/web/docs/project/MOM5_elements.pdf

page X-5 , bottom line, ``typical super-observation coverage'
-- it is not clear what do you by term ``super-observation' (first occurring here, and in caption for Fig. 1, which provides little help in understanding what it is -- is it merely superimposed observational data?).
Please reformulate.

page X-17, , Fig. 3, surface vorticity comparison in top two panels. At first, this figure would be much more informative if both panels have the same minimum and maximum values in its color bars. Secondly, the minimum and maximum values for of vorticity for 10 km plot are ± 0.15 , while for 2.5km resolution they are less ± 0.06 . But isn't it expected that finer resolution simulation should produce larger amplitude vorticity? Or, to be specific, smaller structures and thinner, but more intense filaments? As it appears right now it looks the other way around (although hard to judge because of different color scales). Finally, amplitude of relative vorticity for mesoscale features should be of order of local Coriolis frequency, that is 10^{-4} or something. Especially for anticyclons where relative vorticity has the opposite sign to Coriolis because of centrifugal instability. But the minimum and maximum values on both plots are much much larger. Sure, submesoscale processes can create larger than Coriolis relative vorticity. but what is shown here is unrealistically large. This reviewer believes that this is a mistake.

References

Capet, X., J.~C. McWilliams, M.~J. Molemaker, and A.~F. Shchepetkin, 2008: Mesoscale to submesoscale transition in the California Current System. Part I: Flow structure, eddy flux, and observational tests. *J. Phys. Oceanogr.*, vol. 38, pp. 29-43, doi:10.1175/2007JPO3671.1.

Lemari'e, F., L.~Debreu, G.~Madec, J.~Demange, J.~M. Molinise, and M.~Honnorat, 2015: Stability constraints for oceanic numerical models: implications for the formulation of time and space discretizations. *Ocean Modelling*, vol. 92, pp. 124-148, doi:10.1016/j.ocemod.2015.06.006.

Soufflet, Y., P.~Marchesiello, F.~Lemari'e, J.~Jouanno, X.~Capet, L.~Debreu, and R.~Benshila, 2016: On effective resolution in ocean models. *Ocean Modelling*, vol. 98, pp. 36-50, doi:10.1016/j.ocemod.2015.12.004.

15 March 2017

Response to reviewer's comments

Reviewers' comments:

Reviewer #1 (Remarks to the Author):

Left comments for the Editor only.

Reviewer #2 (Remarks to the Author):

Sub-mesoscale ocean analysis and forecasting

Paul Sandery and Pavel Sakov

The paper examines the results of several Observation System Experiments to demonstrate the degree to which a high resolution numerical model may differ from a lower resolution result. The results are consistent with expectations. The higher resolution model provides more accurate results at scales that are resolved by the observation systems. There are a few points that would preclude acceptance of the paper at this time.

The writing is difficult to follow and understand. The first paragraph of section 3 seems to have lost sentences at the end. The discussion moves from Figure 3 to Figure 4 without pointing out that Figure 4 has become the focus, and the information described in Figure 4 is difficult to discern. I did not find a definition of the statistic MAD used in the presented tables. It is difficult to know what the results mean. Some additional statistics such as root mean squared error, conditional and unconditional bias, etc... would be quite valuable. These are issues that can be addressed easily with careful editing and attention to clear explanation.

Fundamentally there is a disagreement with what is submesoscale. The authors point out that the 10 km model is characterized by mesoscale dynamics and claim that the 2.5 km resolution model contains submesoscale. While the factor of 4 resolution certainly adds physical representation it does not cross from mesoscale to submesoscale. Formal convergence studies of mesoscale dynamics indicate that about 3 km resolution is the point at which numerical methods represent the differential equations involved in mesoscale eddy formation and propagation as measured by eddy kinetic energy levels (Hurlburt and Hogan, 2000). Convergence studies of submesoscale dynamics (Capet et al., 2008) indicate that 1 km resolution is the point at which submesoscale dynamics become represented by numerical methods. Figure 3 shows the vorticity relative to the local Coriolis parameter, and this ratio is a Rossby number. Mesoscale dynamics are characterized by small Rossby number, and submesoscale is reached when the Rossby number becomes order 1 or greater. From the colorbar scale, both these results indicate that the Rossby number is small. Therefore the flow is mainly geostrophically balanced and not reaching into the submesoscale. None of these considerations invalidate the results presented. These can easily be addressed by not referring to the 2.5 km model as submesoscale resolving, and the article title should be changed accordingly.

As computational capability continues to increase, numerical simulations are run at higher resolutions. The observational capability is not keeping pace with the Moore's law increase in computer power. Thus we are reaching the point where numerical models that represent instabilities in the ocean flow field are surpassing the ability of satellites and in situ data to provide information at the modeled scales. Models will generate realistic dynamics, they will generate realistic instabilities in the flow, these instabilities will not be synoptically placed compared to events in the real world, and thus even though numerical models have improved the results are counter intuitively degraded. This is a valuable understanding to bring to light. I think the authors can do this, though there are some issues as pointed out above that are presently obscuring the results and distracting from the main conclusion.

Hurlburt, Harley E., and Patrick J. Hogan. "Impact of 1/8 to 1/64 resolution on Gulf Stream model-data comparisons in basin-scale subtropical Atlantic Ocean models." *Dynamics of Atmospheres and Oceans* 32, no. 3 (2000): 283-329.

Capet, X. A. V. I. E. R., JAMES C. McWilliams, M. JEROEN Molemaker, and A. F. Shchepetkin. "Mesoscale to submesoscale transition in the California Current System. Part I: Flow structure, eddy flux, and observational tests." *Journal of Physical Oceanography* 38, no. 1 (2008): 29-43.

The writing has been improved throughout and the manuscript made easier to understand. The sentence that dropped off from paragraph 1 of section 2 has been fixed. The transition in the discussion between Figure 3 and Figure 4 has been improved and the description of Figure 4 and Figure 4 itself has been improved. A definition and reason for using MAD rather than RMSD has been included. The mean error of forecast innovations we use is the total bias. MAD and bias of a large sample of forecasts and observations are a sufficient indication of system performance.

With regard to the scale of the dynamics we rewrote the manuscript to refer to the 2.5 km system as the higher resolution system and in one place we call it sub-mesoscale permitting. We base this on a new figure (Figure 3), which compares the Rossby number (relative vorticity normalised by planetary vorticity) for the two systems.

We also included references to Hurlburt and Hogan (2000) and Capet et al (2008) in the manuscript, using information contained within these studies to for example support the idea that

predictability projects onto the wide wavenumber spectrum via the inverse energy cascade from the submesoscale dynamics onto the mesoscale dynamics.

Reviewer #3 (Remarks to the Author):

Attached file "review.pdf" is PDF version of this review identical in content. Use this plain text version only as alternative.

Review of manuscript 'Sub-mesoscale ocean analysis and forecasting' by Paul Sandery and Pavel Sakov submitted into Nature Communications

Manuscript Number: NCOMMS-16-14750

Summary and recommendation

This manuscript deals with an emerging dilemma of using a relatively high-resolution model for the purpose of forecast and/or comparison with data available only at a coarser resolution. It is expected that grid refinement leads to improvement of model skill, however, it also results in a more turbulent and, correspondingly, less deterministic on the smaller scales solution which cannot be directly compared to the data (or constrained by the data in the case of data assimilation).

The approach adopted by the authors is to compare filtered model fields (in time in their case) and to use ensemble optimal interpolation which introduces some averaging of non-deterministic component of the solution.

The manuscript is complete -- although too short and somewhat formal (but this is specific to this journal) and well written. It can be published after correcting the items outlined below (some of which I believe are admitted by mistake), and I also suggest de-emphasizing word "sub-mesoscale" because as it used right now it artificially attached to the topic.

We now refer to the 2.5 km resolution system mainly as the 2.5 km system or at least sub-mesoscale permitting. An analysis of Rossby number (Figure 3 in revised manuscript) leads us to believe this. We think sub-mesoscale ocean dynamics are still a useful and relevant reference within the context of the manuscript, which has been rewritten to make clearer the ideas we wish to communicate.

Detailed remarks

The use of term "sub-mesoscale" throughout the article is over-stretched and somewhat misleading. While there is exist a range of definitions of what is considered to be submesocale and what is not (or, more specific to the context of ocean modeling how fine the grid resolution should be for the results to be classified as sub-mesoscale), the common use of word "sub-mesoscale" (like sub-mesoscale processes, etc.) implies strongly ageostrophic dynamics, interactions vortex dynamics with surface boundary layer, formation of fronts and frontal instabilities, convergence zones, etc (c.f. Capet et. al. 2008). And, at the same time much larger scales than the ones at which fill 3D turbulence may occur -- this would be the realm of LES modeling. Somewhat intuitive (but not uniformly accepted) criterion is the requirement that the boundary layer deformation radius should be well resolved by the horizontal grid.

We have reduce the stretching of the term sub-mesoscale by removing from the title and from references to the 2.5 km system. We appreciate the other helpful comments here and have included a reference to Capet et al (2008), which relates the context of our work to this framework.

There is an obvious analogy with the demand to resolve first baroclinic deformation radius for eddy-resolving model, however this analogy is somewhat superficial because the underlying processes are qualitatively different: baroclinic radius generally plays the role of limit to which the eddies are

allowed to grow by mergers, absorptions, and axisymmetrizations -- it is believed that the underlying dynamics is essentially 2D or QG-type turbulence resulting in energy cascade from small to large scales; the sub-mesoscale dynamics is quite different. Besides boundary layer deformation radius is not so easily define. However the de-facto consensus is that sub-mesoscale models require sub-kilometer resolution -- preferably a fraction of Δx .

These are valuable comments and are appreciated. We looked at wave number eddy kinetic energy spectra of the two systems and found the higher resolution model has a longer tail at higher wavenumber as expected and more energy in the mesoscale wave numbers in agreement with Capet et al (2008). This is included in the revised manuscript.

What is actually going on in this study is a B-gridded model derived from MOM4p1 code run at $\Delta x = 2.5$ km, introduced as (page 7, bottom)

"To demonstrate that the 2.5 km system is sub-mesoscale eddy permitting and that the data assimilation and model initialization is not constraining ..."

and compared against $\Delta x = 10$ km simulations using the same code.

While $\Delta x = 2.5$ km is expected produce a healthy vortex dynamics, it is hardly capable to capture sub-mesoscale processes mentioned above.

Especially if it is kept in mind that B-gridded MOM4p1 and MOM5 codes use second-order-accurate vector-invariant form (based on absolute vorticity and Bernoulli potential) for discretization of momentum equations.

The use of B-grid is generally considered a kind of " $\sqrt{2}$ " disadvantage in comparison with a C-gridded model also using vector-invariant form and second-order discretization, e.g., MITgcm and NEMO [see, for example Griffies own Elements MOM (2012 release) Chapters 9 and 14; web link provided below in this review].

Furthermore, ROMS uses 3rd-order upstream-biased advection for the momentum equation, which is known to offer a significant advantage over the second-order numerics for advection of momenta (Lemarié et al., 2015; Soufflet et al., 2016 among others), which pushes threshold for submesoscale resolution for MOM4p1 code even further

(it should be noted that most submesoscale simulations to date were produced by ROMS or by pseudo-spectral codes in idealized studies). Nevertheless, this reviewer believes that the methodological approach of comparing 10 km and 2.5 km models for the purpose of making the point that

"... whilst forecast errors are larger in the sub-mesoscale system, the system can nonetheless reproduce detailed dynamical features observed in satellite remote sensing products such as AVHRR SST and MODIS chlorophyll-imagery... and..." is actually correct and worth making. The authors should just remove overemphasis on submesoscale (perhaps change wording of the title): after all what going from 10 to 2.5 km resolution results in solid improvement of model skill, and the methodology of comparing filtered model fields against the available data (rather than attempting a more direct comparison) is perfectly meaningful way to judge the model accuracy and errors.

Thanks for that, the title and wording has been changed to reflect these comments.

page X-3, 1/3 from bottom, sentence "Higher-resolution models should theoretically provide improved dynamical solutions. This is because they resolve more and parameterize less of the subgrid-scale physics" -- actually this needs to be clarified or reformulates. Parameterization usually involves a physically-motivated component of the code designed to represent processes which ocean model cannot simulate directly due to lack of resolution or approximations made to fundamental equations or both. In principle this means that once grid resolution is refined, the modeler may choose to drop some of these components with the expectation that the processes they are designed to represent are now modeled directly. However, reading further into the article does not seem to support this -- what is going on there is just running exactly the same code at higher resolution.

Please reformulate.

After the above quote we have added

'When downscaling there remains, however, unresolved physics requiring recalibration of sub-grid scale parameterisations.'

Later we now write

'Both systems are run with identical settings except for resolution and the scale dependent Smagorinsky horizontal mixing scheme. This aims to take care of recalibration of sub-grid scale horizontal mixing for the downscaled system, which is considered to be the most important parameterisation requiring tuning for a change in resolution.'

\bf page X-5, 1/3 from bottom, reference [Griffes et al., 2009]
-- the reference is essentially a manual, which is updated and expanded on regular basis. Why to refer to an outdated document when a newer and more complete is available?

http://www.mom-ocean.org/web/docs/project/MOM5_elements.pdf

We refer to the MOM4p1 manual as this is the version of the code used in the study.

page X-5 , bottom line, ``typical super-observation coverage'
-- it is not clear what do you by term ``super-observation' (first occurring here, and in caption for Fig. 1, which provides little help in understanding what it is -- is it merely superimposed observational data?).
Please reformulate.

We have added the following and a reference to explain super-observations

'A super-observation is formed by combining all observations within a model grid cell into a single observation weighted by their inverse error variance. The super-observation therefore places more emphasis on observations with lower error estimates. The super-observation also has a new error estimate that is lower than the original observations, thereby increasing the impact of the observations in the assimilation. This is standard practice in weather and ocean forecasting and reduces discrepancies related to representation error (Oke and Sakov, 2008)'

page X-17, , Fig. 3, surface vorticity comparison in top two panels. At first, this figure would be much more informative if both panels have the same minimum and maximum values in its color bars. Secondly, the minimum and maximum values for of vorticity for 10 km plot are ± 0.15 , while for 2.5km resolution they are less ± 0.06 . But isn't it expected that finer resolution simulation should produce larger amplitude vorticity? Or, to be specific, smaller structures and thinner, but more intense filaments? As it appears right now it looks the other way around (although hard to judge because of different color scales). Finally, amplitude of relative vorticity for mesoscale features should be of order of local Coriolis frequency, that is 10^{-4} or something. Especially for anticyclons where relative vorticity has the opposite sign to Coriolis because of centrifugal instability. But the minimum and maximum values on both plots are much much larger. Sure, submesoscale processes can create larger than Coriolis relative vorticity. but what is shown here is unrealistically large. This reviewer believes that this is a mistake.

These issues have been addressed by the introduction of Figure 3, which corrects an error in the scales present in the original plot. Figure 3 also now deals with the relative importance of vorticity and planetary vorticity, which is essentially a Rossby number analysis. We have also added a reference to McWilliams (2016) as the same analysis was done in this article.

References

Capet, X., J.-C. McWilliams, M.-J. Molemaker, and A.-F. Shchepetkin, 2008: Mesoscale to submesoscale transition in the California Current System. Part I: Flow structure, eddy flux, and observational tests. *J. Phys. Oceanogr.*, vol. 38, pp. 29-43, doi:10.1175/2007JPO3671.1.

Lemarié, F., L.-Debreu, G.-Madec, J.-Demange, J.-M. Molinèse, and M.-Honnorat, 2015: Stability constraints for oceanic numerical models: implications for the formulation of time and space discretizations. *Ocean Modelling*, vol. 92, pp. 124-148, doi:10.1016/j.ocemod.2015.06.006.

Soufflet, Y., P.-Marchesiello, F.-Lemarié, J.-Jouanno, X.-Capet, L.-Debreu, and R.-Benshila, 2016: On effective resolution in ocean models. *Ocean Modelling*, vol. 98, pp. 36-50, doi:10.1016/j.ocemod.2015.12.004.

Reviewer #4 (Remarks to the Author):

The paper provides a study of the forecasting skill of an ocean model with assimilation when its resolution is increased. The main finding is that - somehow counterintuitively - the increase in resolution lowers in general the mesoscale forecasting skill, although in some cases fine scale features (like in sea surface temperature) compare more favorably to remote sensed observations. The paper is in general well written and the numerical experiment appears to me well conceived. Considering the advancement in high resolution modelling and the interest in applying higher resolution schemes for forecasting I find the theme of general interest and timely. The authors in my opinion have also addressed most of the remarks raised by the reviewers. Some issues (a few of them important) however remain, which I list below and which are aimed either at strengthening the result, or at reaching a potentially larger community of non-specialists.

1. Considering that the higher resolution model is more turbulent and hence less deterministic than the lower resolution one, and that observations for constraining the finer scales are not available, the result of the paper is surprising only at first sight and in fact is probably in line with expectations for high resolution modelers. Therefore, I think that besides describing in details the results (which is important and well achieved in the current manuscript), the authors should also present some consequences and some possible ways forward for fine scale forecasting. Although I understand that the format of the journal probably imposes some limitations, I find that a few final concise sentences more would greatly help specialists and non-specialists in appreciating more the manuscript. The last sentence of the paper claiming generic "broad implications" is to me by far too generic and should be extended. One important point - which has been mentioned before and which may attract interest from the observational community - is that computer power has grown faster than the capacity of satellite and in situ observing systems to provide information at the modelled scales. As a consequence, the finer scales produced by the models, even when dynamically realistic, are not placed individually as in reality. Other comments of general interest are also welcome.

2. Innovation metric: the result presented is very dependent on the performance quantification. The authors claim that the metric they use, the Mean Absolute Deviation (MAD), is better suited to their end, even if the most common one is the Root Mean Square Deviation (RMSD). The authors provide a short argument for their choice, but given the importance of the choice, and the qualitative nature of their argument, I feel that some references are needed as well. If references are not available about this point, further analysis in the Supplementary Information may be possibly needed to make their argument more quantitative. I would also like to know whether the result still holds if MAD is used instead of RMSD.

3. When considering a paper which is based on a negative result - in this case, the incapacity of a higher resolution model to improve mesoscale forecasting skills - there is the question of whether the paper describes an intrinsic limit, or whether the negative result is due to the fact that the experiment has not been optimized enough. In this case, the result is in line with expectations, and methodologically, my guess is that the model is used with a standard configuration that does not provide much room for tuning. Maybe this point is even obvious for a specialist in ocean assimilation. As a non-specialist reader, I would like however to be reassured by the authors that the model configuration is standard and that the results is not likely to change by optimising it.

4. The text is in general well written, but it should be polished more (some sentences are difficult to follow; there is an "and and" typo).

Response to reviewers comments

Reviewer #4 (Remarks to the Author):

The paper provides a study of the forecasting skill of an ocean model with assimilation when its resolution is increased. The main finding is that - somehow counterintuitively - the increase in resolution lowers in general the mesoscale forecasting skill, although in some cases fine scale features (like in sea surface temperature) compare more favorably to remote sensed observations. The paper is in general well written and the numerical experiment appears to me well conceived. Considering the advancement in high resolution modelling and the interest in applying higher resolution schemes for forecasting I find the theme of general interest and timely. The authors in my opinion have also addressed most of the remarks raised by the reviewers. Some issues (a few of them important) however remain, which I list below and which are aimed either at strengthening the result, or at reaching a potentially larger community of non-specialists.

1. Considering that the higher resolution model is more turbulent and hence less deterministic than the lower resolution one, and that observations for constraining the finer scales are not available, the result of the paper is surprising only at first sight and in fact is probably in line with expectations for high resolution modelers. Therefore, I think that besides describing in details the results (which is important and well achieved in the current manuscript), the authors should also present some consequences and some possible ways forward for fine scale forecasting. Although I understand that the format of the journal probably imposes some limitations, I find that a few final concise sentences more would greatly help specialists and non-specialists in appreciating more the manuscript. The last sentence of the paper claiming generic "broad implications" is to me by far too generic and should be extended. One important point - which has been mentioned before and which may attract interest from the observational community - is that computer power has grown faster than the capacity of satellite and in situ observing systems to provide information at the modelled scales. As a consequence, the finer scales produced by the models, even when dynamically realistic, are not placed individually as in reality. Other comments of general interest are also welcome.

We thank the reviewer for the helpful comments. It is difficult to describe consequences and possible ways forward for fine scale forecasting given there are many details and this is an emerging area of research. We have replaced the final statement claiming 'broad implications' with the following:

"The growth in computer power has enabled increases in model resolution and more realistic representation of dynamics and variability (Chassignet and Xu, 2017). We found that an increase in resolution doesn't automatically improve the skill

of the forecasting system. Sub-mesoscale ocean forecasting certainly also requires better observation coverage, improved physics and sub-grid scale parameterisations.”

2. Innovation metric: the result presented is very dependent on the performance quantification. The authors claim that the metric they use, the Mean Absolute Deviation (MAD), is better suited to their end, even if the most common one is the Root Mean Square Deviation (RMSD). The authors provide a short argument for their choice, but given the importance of the choice, and the qualitative nature of their argument, I feel that some references are needed as well. If references are not available about this point, further analysis in the Supplementary Information may be possibly needed to make their argument more quantitative. I would also like to know whether the result still holds if MAD is used instead of RMSD.

The very use of MAD rather than RMSD is deliberate. We refer the reviewer to Sakov and Sandery 2014 Appendix A for the justification and have replaced the existing paragraph

“The performance of both systems is quantified in terms of forecast innovation error using all available forward and independent observations. The most common innovation metric is root mean square deviation (RMSD), which can be dominated by a relatively small number of innovation elements with large magnitudes corresponding to either less observed or more chaotic parts of the model, or be influenced by observations with large errors as it does not account for observation error. Consequently it may not accurately represent the performance of the system. To minimise this we use Mean Absolute Deviation (MAD).”

with

“The performance of both systems is quantified in terms of forecast innovation error using all available forward and independent observations. The most common innovation metric is root mean square deviation (RMSD), which can be dominated by a relatively small number of innovation elements with large magnitudes corresponding to either less observed or more chaotic parts of the model, or be influenced by observations with large errors as it does not account for observation error. Consequently it may not accurately represent the performance of the system. To minimise this we use Mean Absolute Deviation (MAD). For information illustrating that MAD is a more robust metric than RMSD see Huber and Ronchetti [2009]. An example related to assessing performance of ocean forecasting systems can be found in Appendix A of Sakov and Sandery [2015].”

We also provide the following supplementary figures as examples. Our conclusions do not change if we use RMSD, however, they are more robust using MAD. If error distribution is Gaussian then RMSD is 20% higher than MAD - this is in general the case for SST, however, note the spike in RMSD at cycle 145 in SLA due to non-gaussian error distribution. This could offset the mean RMSD for SLA in a misleading way.

Figure 1. Innovation errors for the 10 km system for the three year reanalysis from 2006-2009. Cycle length is 3 days. The two relevant metrics for comparison here are FMAD - Forecast Mean Absolute Deviation and FRMSD - Forecast Root Mean Squared Deviation.

3. When considering a paper which is based on a negative result - in this case, the incapacity of a higher resolution model to improve mesoscale forecasting skills - there is the question of whether the paper describes an intrinsic limit, or whether the negative result is due to the fact that the experiment has not been optimised enough. In this case, the result is in line with expectations, and methodologically, my guess is that the model is used with a standard configuration that does not provide much room for tuning. Maybe this point is even obvious for a specialist in ocean assimilation. As a non-specialist reader, I would like however to be reassured by the authors that the model configuration is standard and that the results is not likely to change by optimising it.

We have added the following statements in the conclusions section regarding this;

“The results are based on standard configurations of ocean model and data assimilation system. The two experiments utilise the same settings apart from differences in resolution. These results point to the existence of an intrinsic limit and the incapacity of a higher resolution model to improve forecasting of the mesoscale circulation.”

4. The text is in general well written, but it should be polished more (some sentences are difficult to follow; there is an "and and" typo).

The typo has been corrected

The following sentence was removed as it was not easy for the reader to see this

“Looking at 18th November 2008 it’s seen that SLA and SST from the 2.5 km system appears to agree best with the MODIS and AVHRR data”

The paragraph in the Introduction beginning with “In this study Ensemble Optimal Interpolation is used.....“ was moved into the Methods section, which was moved to be the last section as per the guidelines.

Additional amendments

Black text - original

Blue text - changed text

“Ocean modelling simulations of the Gulf Stream at 1/64° horizontal resolution were shown to produce more realistic patterns than equivalent lower resolution simulations [Hurlburt and Hogan, 2000]”

“Ocean modelling simulations of the Gulf Stream at submesoscale resolving horizontal resolution have been shown to produce more realistic patterns than equivalent lower resolution simulations [Hurlburt and Hogan, 2000; Chassignet and Xu, 2017]”

“Both systems are run with identical settings except for resolution and the scale dependent Smagorinsky horizontal mixing scheme. This aims to take care of recalibration of sub-grid scale mixing for the downscaled system, which is considered to be the most important parameterisation requiring tuning for a change in resolution.

“Both systems are run with identical settings except for grid resolution. The scale dependent Smagorinsky horizontal mixing scheme is used to take care of recalibration of sub-grid scale mixing for the change in resolution. “

“EnOI requires a background static ensemble of model error covariances, in this case this ensemble is derived from a long (18 year) nature hindcast run of an eddy resolving global model.”

“We use a stationary ensemble of model error covariances derived from an 18 year nature run of an eddy resolving global model.”

The most common innovation metric is root mean square deviation (RMSD)

A common innovation metric is root mean square deviation (RMSD)

“There are days such as 23rd May 2007 and the 15th September 2007 where the SLA fields from the four systems are in approximate agreement between systems. The predictions by the four systems on these occasions are also in reasonable agreement with the MODIS chlorophyll- α data.”

“There are days such as 23rd May 2007 and 15th September 2007 where SLA fields from the four systems are in reasonable agreement with each other and with the MODIS chlorophyll- α data.”

“When we look at the entire reanalysis this way, however, we see that the 2.5 km agrees less of the time with the satellite imagery compared to the 10 km systems.”

“When we look at the **En100** reanalysis this way, which is not shown here, we see that the 2.5 km agrees less of the time with the satellite imagery compared to the 10 km systems.”

References added to manuscript

Chassignet, E. P., and X. Xu (2017), Impact of horizontal resolution (1/12 to 1/50) on gulf stream separation, penetration, and variability, *Journal of Physical Oceanography*, (2017).

Huber, P. J., and E. M. Ronchetti (2009), *Generalities - Robust Statistics*, pp. 1–21, John Wiley & Sons, Inc.

Sakov, P., and P. Sandery (2015), Comparison of EnKF and EnOI regional ocean reanalysis systems, *Ocean Modelling* 89, 45-60.